# Gallium-Doped Hydroxyapatite: Shape Transformation and Osteogenesis Activity

**DOI:** 10.3390/molecules28217379

**Published:** 2023-11-01

**Authors:** Wei Shuai, Jianguo Zhou, Chen Xia, Sirui Huang, Jie Yang, Lin Liu, Hui Yang

**Affiliations:** 1School of Rehabilitation Medicine, Gannan Medical University, Ganzhou 341000, China; shuaiwei@gmu.cn; 2Key Laboratory of Biomaterials and Bio-Fabrication in Tissue Engineering of Jiangxi Province, Ganzhou 341000, China; huangsirui@gmu.cn (S.H.); 13419472246@163.com (J.Y.); 3Department of Joint Surgery, Ganzhou People’s Hospital, Ganzhou 341000, China; zjg840818@163.com; 4Sichuan Volcational College of Cultural Industries, Chengdu 610213, China; chenxia0810@163.com; 5School of Medical Information Engineering, Gannan Medical University, Ganzhou 341000, China

**Keywords:** gallium, hydroxyapatite, osteogenesis

## Abstract

In this study, we employed a chemical precipitation method to successfully synthesize nanoparticles of gallium-doped hydroxyapatite (Ga-HAp). The microstructure of Ga-HAp was precisely tailored by modulating the concentration of gallium ions. Our findings unequivocally demonstrate that gallium ions exert a pronounced inhibitory influence on the growth of HAp crystals, and this inhibitory potency exhibits a direct correlation with the concentration of gallium. Furthermore, gallium ions facilitate the metamorphosis of HAp nanoparticles, transitioning them from nanoneedles to nanosheets. It is worth noting, however, that gallium ions exhibit a limited capacity to substitute for calcium ions within the crystal lattice of HAp, with the maximum substitution rate capped at 4.85%. Additionally, gallium plays a pivotal role in constraining the release of ions from HAp, and this behavior remains consistent across samples with varying Ga doping concentrations. Our in vitro experiments confirm that Ga-doped HAp amplifies both the proliferation and osteogenic differentiation of bone marrow mesenchymal stem cells.

## 1. Introduction

Natural bone tissue possesses a hierarchical structure characterized by the precise arrangement of inorganic minerals and an organic matrix. This intricate configuration imparts robust mechanical properties to bone tissue, ensuring the protection of human tissues and enabling essential mobility functions [1]. Although bone tissue exhibits innate self-repair capabilities, surgical implantation of bone or bone-like tissue becomes imperative when bone defects exceed a critical size, thereby promoting effective bone tissue regeneration [2,3].

Biomimetic ceramics, particularly hydroxyapatite (HAp), are widely employed for bone regeneration due to their chemical composition resembling that of natural bone minerals [4,5,6]. This similarity grants HAp exceptional biocompatibility, bioactivity, and osteogenic potential, which has garnered significant interest. However, pure-phase HAp faces challenges in bone repair, including a slow degradation rate and limited osteoinductive activity [7]. Furthermore, it struggles to address the demands of repairing bone defects arising from pathological conditions such as infections, tumors, and osteoporosis [8]. While achieving these functionalities through drug loading is conceivable, challenges remain in terms of restricted drug loading capacity, intricate control of drug release behavior, and achieving sustained drug release [9,10].

In bone tissue, a diverse array of trace elements, including Mg, Zn, Sr, Fe, Mn, Ga, and even rare earth elements, play pivotal roles in the intricate process of bone regeneration. Typically, calcium ions within the hydroxyapatite structure readily undergo substitution with other metal ions, resulting in the creation of numerous defects. Consequently, this leads to a reduction in crystallinity coupled with an increase in solubility, ultimately culminating in the substantial release of bioactive ions in localized regions, thereby amplifying the material’s osteogenic and angiogenic activity. Extensive research is thus devoted to enhancing the bone repair performance of hydroxyapatite through ion doping [11,12].

Gallium (Ga), a metallic element discovered by chemist Paul in 1875, is positioned in the third main group of the periodic table. Its pronounced affinity for tissue growth and remodeling, including bone and tumor tissues, has been elucidated through extensive research [13]. Within the realm of bone tissue, Ga’s potential for anti-resorptive effects has garnered significant attention [14]. Clinical applications of Ga can be categorized into two types: firstly, the suppression of bone resorption and associated pain caused by multiple myeloma and bone metastasis [15]; secondly, the addressing of hypercalcemia induced by malignant tumors [16]. Furthermore, Ga finds application in treating osteitis deformans (Paget’s disease of bone) [17]. In vitro studies have revealed that Ga can effectively inhibit osteoclast differentiation and bone resorptive activity without affecting osteoblasts, thus mitigating the risk of bone fractures. Deeper investigations have demonstrated that Ga achieves the suppression of osteoclast differentiation by inhibiting NFATc1 expression [18]. Osteoporosis, characterized by diminished bone density and degradation of bone microstructure, resulting in reduced bone toughness and heightened fracture susceptibility, represents a systemic ailment. Bisphosphonates have demonstrated commendable effectiveness in elevating bone density, curbing bone loss, and diminishing fracture risk within clinical practice. However, their requirement for long-term, regular medication administration poses considerations [19,20].

Moreover, numerous studies have highlighted Ga’s capacity to impede the nucleation and growth of HAp crystals [21]. Surprisingly, scant attention has been paid in the literature to the impact of Ga on the microscopic morphology of HAp nanoparticles, despite the profound connection between particle topology and the osteogenic differentiation behavior of mesenchymal stem cells. Furthermore, Ga, being a trivalent metal ion, engenders ongoing debate regarding its potential for high-dose, stable doping within HAp crystals in mild synthesis systems.

Given Ga’s promising osteogenic activity and its affinity with HAp crystals, we successfully synthesized Ga-doped HAp nanoparticles via the chemical precipitation method. Our study comprehensively investigated the influence of Ga incorporation on the crystal structure, morphology, and phase composition of HAp crystals. Furthermore, we assessed the potential effects of Ga-doped HAp powder and bulk materials on the proliferation and osteogenic differentiation behavior of bone marrow mesenchymal stem cells.

## 2. Results and Discussion

In this study, Ga-doped hydroxyapatite (Figure 1) was synthesized using the chemical precipitation method (Figure 1). In the context of synthesizing HAp, the incorporation of additional ions not only exerts an influence on the crystallinity and lattice structure of Hap, but also plays a role in modulating the aggregation behavior of nanocrystals [12,22]. Consequently, distinctive micro–nano morphologies emerge in the final product. In the present investigation, the introduction of Ga into the HAp synthesis process was pursued. Examination of SEM images unveiled the fact that Ga integration could effectively mitigate the inherent self-aggregation tendencies of nanocrystals (Figure 1). Moreover, research studies have indicated that Ga has the capacity to hinder the growth of HAp crystals, resulting in the formation of smaller particles [23,24]. However, it is important to note that this particular conclusion cannot be directly inferred from Figure 1. It is noteworthy that while the extent of Ga doping is proportional to the concentration of Ga in the reaction solution, the ultimate quantity of Ga incorporated into the crystal lattice of HAp remains relatively modest. In the case of 10 Ga-HAp, the Ga doping level stands at a mere 0.76% (in atomic percentage). Even in 50 Ga-HAp, the doping level only reaches 4.85%. This observation points to the challenge Ga encounters in effectively integrating into the crystal lattice of HAp.

To further validate the SEM results, we conducted detailed examinations of the product morphology using TEM (Figure 2). From the TEM results, it can be observed that when Ga was not present in the reaction system, the product primarily consisted of nanorod-shaped particles with a length of approximately 100 nm (Figure 2A,B). Additionally, scattered nanosheet-like particles were also observed within the field of view. The surface of these nanosheet structures exhibited an ordered stacking arrangement of nanorod-shaped particles in the two-dimensional direction. At a Ga doping concentration of 10%, the length of the nanorod-shaped particles shortened, and their quantity decreased relatively, indicating that the introduction of Ga inhibited the growth of nanorod-shaped particles along their long axis (Figure 2C,D). When the Ga doping level reached 25%, the product consisted mainly of nanosheet-like structures, although isolated nanorod-shaped particles could still be observed (Figure 2E,F). With a further increase in Ga doping to 50%, the product consisted predominantly of nanosheet-like particles (Figure 2G,H). However, it is worth noting that while the size and quantity of nanorod-shaped particles changed significantly with increasing Ga content, the size of nanosheet-like particles remained relatively unchanged. In a word, Ga incorporation was found to inhibit the growth of nanorod-shaped units, particularly along their long axis, while having minimal impact on the assembly of nanorod-shaped particles in the two-dimensional direction. This ultimately resulted in a shift from nanorod-shaped particles being the predominant form to nanosheet-like structures becoming predominant as the Ga content increased.

We conducted XRD analysis to investigate the crystal phase of the product (Figure 3). The results revealed that no other crystal phases were present in the product apart from the HAp crystal phase, indicating that the introduction of Ga did not lead to the formation of other crystal phases. Additionally, it is noteworthy that previous research has reported Ga’s ability to inhibit the nucleation and growth of HAp crystals [25]. Therefore, relative to the HAp control samples, the introduction of Ga should result in a reduction in crystallinity, manifesting as broadening of XRD peaks. The crystalline peak at around 25° corresponds to the (002) crystal plane diffraction of HAp. It can be observed from the XRD pattern that with the introduction of Ga, the relative intensity of the (002) crystal plane diffraction peak decreased. Hence, we can deduce that Ga can inhibit the preferential growth of HAp along the c-axis direction. Furthermore, as the amount of introduced Ga increased, the (002) crystal plane exhibited a shift towards higher angles. This observation suggests that the incorporation of Ga induces a contraction in the unit cell volume of hydroxyapatite. This phenomenon can be attributed to the smaller ionic radius of Ga (0.062 nm) in comparison to Ca (0.099 nm). Notably, this outcome corroborates findings reported in other studies [26]. Combining these results with the TEM analysis, it is apparent that during the synthesis process, HAp tends to spontaneously grow along the c-axis, leading to the formation of rod-shaped nanocrystals, with the long axis of the nanorods aligned along the c-axis. However, the introduction of Ga suppresses the growth of nanorods along their long axis, thereby inhibiting their c-axis-oriented growth tendency. As the Ga content increases, this inhibitory effect becomes more pronounced [23]. Furthermore, the results of the FTIR analysis indicate that the introduction of Ga does not lead to significant changes in the chemical composition of the product (Figure 4). Simultaneously, the results reveal distinct carbonate absorption peaks in the infrared absorption spectrum. These carbonate ions might originate from the hydration of carbon dioxide in the air during the synthesis process, leading to the formation of carbonate ions. These carbonate ions then enter the HAp lattice, resulting in the formation of carbonated hydroxyapatite. This phenomenon has been extensively reported in the literature [27,28].

The Ca and Ga ion-release profiles of the products before and after Ga^3+^ doping are shown in Figure 5. The results indicate that the doping of Ga ions significantly reduces the release level of Ca ions in pure HAp. After 14 days of release, the Ca ion release from pure-phase HAp particles is approximately 700 μg, while after Ga^3+^ doping, the Ca ion release from the product decreases to around 400 μg, indicating that Ga ion doping can significantly inhibit the dissolution of HAp crystals. This observation aligns with findings reported in other literature [29], supporting the clinical use of Ga to treat conditions such as hypercalcemia caused by bone cancer [30]. Upon ingestion of soluble gallium salts by the body, Ga deposits within bone mineral tissue, inhibiting mineral dissolution behavior and consequently lowering the blood Ca^2+^ levels [31]. It is noteworthy that, with increasing Ga doping levels, there are no significant differences observed in the release levels of either Ca^2+^ or Ga^3+^ among the three groups of doped samples. This could be attributed to the overall higher levels of Ga doping, resulting in similar dissolution behaviors in the doped samples.

Research has revealed the potential cytotoxicity of micro–nanoparticles [32,33]. To validate the biocompatibility of Ga-doped HAp particles, we investigated cell proliferation after co-culturing the powder with cells (Figure 6A). The results demonstrate that pure-phase HAp exhibits an inhibitory effect on cell proliferation, while this inhibitory effect diminishes upon Ga doping of HAp. This phenomenon might be attributed to the potential of high concentrations of Ca^2+^ released by low-crystallinity HAp to inhibit cell proliferation. Ga doping, however, lowers the release levels of Ca^2+^, thereby reducing the inhibitory effect of the powder on cell proliferation.

Subsequently, we further examined cell proliferation behavior on the surface of bulk materials (Figure 6B). The results demonstrate that both HAp and Ga-doped HAp exhibit excellent biocompatibility. After 1 day of cultivation, cells on the material surface maintain good activity. As the cultivation period extends to 5 days, Ga-doped samples are more conducive to cell proliferation compared to the undoped samples. However, no significant differences are observed among the 10 Ga-HAp, 20 Ga-HAp, and 50 Ga-HAp groups. Studies have indicated that under in vitro culture conditions, a certain concentration of Ca^2+^ can promote osteoblast proliferation, while high concentrations of Ca^2+^ can exhibit inhibitory effects on cell proliferation [34]. The lower calcium ion release rate of the Ga-doped materials compared to undoped HAp might be the direct reason for the enhanced cell proliferation behavior, as evident from the ion release results [35]. Furthermore, due to the similar ion release behaviors among the three Ga-doped groups, no significant differences are detected among them.

As the ion release behavior and biocompatibility of the three groups of Ga-doped HAp samples were comparable, we selected 10 Ga-HAp for further investigation of its osteogenic activity, with GaOOH used as the control group (Figure 7). HAp significantly upregulates the expression of the ALP gene, and this enhancing effect is well maintained, even after extending the cultivation time to 14 days. However, when Ga is introduced into the material, the promotion of ALP gene expression by 10 Ga-HAp is less pronounced. Similarly, HAp promotes the expression of the ColI gene in osteoblasts, but the modified materials exhibit a certain degree of attenuation in upregulating cell osteogenic differentiation compared to pure HAp. Notably, at 14 days of induction culture, there is a significant difference between the 10 Ga-HAp group and the control group, but the difference compared to the HAp is not as apparent. The expression peak of the OC gene during osteogenic differentiation corresponds to the mature stage of cells, where a large number of calcium nodules appear in the culture environment, facilitating new bone formation. The results indicate that pure-phase HAp is most effective in upregulating OC gene expression, and after 14 days of culture, both the HAp and 10 Ga-HAp groups can promote OC gene expression.

Ca^2+^ is ubiquitous in living organisms and is closely associated with various cellular processes such as fertilization, mitosis, neurotransmission, muscle contraction and relaxation, gene transcription, and cell death. Generally, the intracellular calcium ion concentration is around 100 nM, but it can increase to 500–1000 nM under external stimuli [36]. Additionally, different calcium ion concentrations have a significant impact on osteoblast behavior. Shinichi et al. [35], using type II collagen loaded with varying concentrations of Ca^2+^ as a model, studied the effects of different Ca^2+^ concentrations on osteoblasts. They found that calcium ion concentrations of 2–4 mM were conducive to maintaining cell viability and promoting cell proliferation, while an environment with 6–8 mM Ca^2+^ favored osteoblast differentiation. However, when calcium ion concentration exceeded 10 mM, it led to some cytotoxicity. Another study by Ekolou-Kalonji et al. [37] revealed that when calcium ion concentration was between 2 and 5 mM, there were no significant changes in cell proliferation behavior, but differentiation rates decreased with higher calcium ion concentrations. Further increasing calcium ion concentration to 7–10 mM resulted in a decrease in both osteoblast proliferation and differentiation behavior.

In this study, both HAp and Ga-HAp continuously released Ca^2+^ during cell culture, participating in various cellular processes. The introduction of Ga led to a decrease in the rate of ion dissolution from the material, consequently resulting in a relatively lower calcium ion concentration in the culture medium compared to the HAp group. This might explain the differences in the promotion of osteogenic gene expression levels between the two materials. The consistent expression of the OC gene in the later stages of culture suggests that as the culture time extends the expression of the OC gene becomes less sensitive to changes in Ca^2+^ concentration within a certain range.

## 3. Materials and Methods

### 3.1. Preparation of Ga-Doped HAp

A total of 10 mL of 0.1 mol/L Ga(NO_3_)_3_ solution and 90 mL of 0.1 mol/L Ca(NO_3_)_2_ solution were mixed, and the mixture was continuously stirred for 30 min. Subsequently, 60 mL of 0.1 mol/L (NH_4_)_2_HPO_4_ solution was added dropwise to the above mixture, while maintaining high-speed stirring during the reaction. After 30 min of reaction, the pH of the solution was adjusted to 10 using ammonia solution. Following 24 h of stirring at room temperature, the precipitate particles were washed three times with deionized water, separated by centrifugation, and then transferred to a 50 °C oven for 24 h of drying, resulting in Ga-doped HAp. According to the molar ratio of Ga/(Ga+Ca), the sample was labeled as 10 Ga-HAp. Additionally, we obtained HAp, 20 Ga-HAp, and 50 Ga-HAp samples by varying the content of Ga and utilizing the synthesis method described above.

### 3.2. Material Characterization

The phase composition of the inorganic powders was analyzed using an X-ray diffraction spectrometer (XRD; X’Pert PRO, PANalytical Co., Amsterdam, The Netherlands) with a working voltage of 40 kV and current of 40 mA. The scanning range was 10° to 80°, and the scanning speed was 2°/min. Scanning electron microscopy (SEM, Carl Zeiss Sigma NTS Gmbh, Oberkochen, Germany) and transmission electron microscopy (TEM; FEI, Tecnai G220, Oregon, OR, USA) were employed to observe the microstructure and morphology of the products. The chemical composition of the products was analyzed using Fourier-transform infrared spectroscopy (FTIR; Nicolet IS 10, ThermoFisher Scientific, Waltham, MA, USA), with a spectral scanning range of 4000 to 400 cm^−1^.

### 3.3. Cumalative Release Profile of Ca and Ga Ions from the Powders

We weighed 40 mg of powder and placed it in 20 mL of PBS solution. Subsequently, we centrifuged it at specific time points, collected 5 mL of the supernatant, and replaced with an equal volume of fresh PBS solution to mix in the release system. We measured the concentration of calcium ions (Ca^2+^) and gallium ions (Ga^3+^) in the collected samples using inductively coupled plasma mass spectrometry (ICP-MS, NexION 350X, PerkinElmer, Shelton, CT, USA).

### 3.4. Cell Culture

Mouse bone marrow mesenchymal stem cells (BMSCs, CRL-12424, ATCC) were utilized to investigate the cytocompatibility and osteogenic activity of the powder particles and bulk materials. The cells were cultured in DMEM medium containing 10% fetal bovine serum, 100 IU/mL penicillin, and 0.1 mg/mL streptomycin. The culture environment was maintained at 5% carbon dioxide and 37 °C with humidity. The culture medium was changed every two days during cell cultivation. When the cell confluence reached 80–90%, the cells were detached using 0.25% trypsin-EDTA solution and centrifuged at 125× *g* to separate the cells. The cells used in this experiment were from the 5th passage.

### 3.5. Biocompatibility Evaluation of the Powder and Bulk Materials

Cells were seeded into 24-well plates at a density of 1 × 10^4^ cells per well, with 1 mL of culture medium added to each well, and then incubated overnight. Sterile powder was dispersed in the cell culture medium at concentrations of 200 μg/mL. After co-culturing the cells with the powders for 1, 3, and 5 days, the cell numbers were determined using the CCK-8 assay kit to assess cell proliferation status in the co-culture system.

Simultaneously, we assessed the biocompatibility of the bulk material products. A quantity of 0.2 g of powder was poured into a mold, and the mold was subjected to pressure at 4 MPa for 5 min. After pressure release, the circular samples (1 cm in diameter) were removed from the mold, soaked in 75% ethanol overnight, and then washed three times with PBS for further use. Additionally, polystyrene discs with a diameter of 1 cm were used as a control group. Before the bulk materials were transferred to a 24-well plate, the well was treated with agarose, and then 20 μL of cell suspension containing 5 × 10^3^ cells was dropped onto the surface of the bulk materials. After incubating for 2 h, 1 mL of culture medium was added. The CCK-8 assay kit was used to measure cell proliferation on the co-cultured bulk materials for 1, 3, and 5 days.

### 3.6. In Vitro Effects of Ga-Doped HAp Powders and Bulk on Osteoblastogenesis

A total of 2 × 10^4^ cells were seeded onto the surface of the bulk materials, and GaOOH, prepared using the coprecipitation method, was used as the control group. After 1 day of incubation in growth medium, the osteogenic induction medium (RASMX-90021, Cyagen Biosciences, Inc., Santa Clara, CA, USA) was substituted for the growth medium. This was replaced with fresh medium every 2 days. After co-culturing for 14 days, the expression levels of osteogenic-related genes such as type I collagen (ColI), alkaline phosphatase (ALP), and osteocalcin (OC) were detected using real-time polymerase chain reaction (RT-PCR) technology.

### 3.7. Statistical Analysis

In this study, experimental data were presented as the mean ± standard deviation (mean ± SD). One-way analysis of variance (ANOVA) and Tukey’s multiple comparison test were performed using SPSS software to analyze the data. Statistical significance was confirmed when the *p*-value was less than 0.05.

## 4. Conclusions

In conclusion, this study successfully synthesized highly concentrated Ga-doped HAp material. The introduction of Ga directly influenced the growth behavior of HAp crystals, inhibiting their oriented growth along certain directions. With increasing Ga doping content, the morphology of the particles transitioned from nanoneedles to nanosheets. Additionally, Ga incorporation significantly reduced the dissolution rate of HAp crystals, thereby enhancing the material’s biocompatibility and promoting the proliferation of mBMSCs. In comparison to pure HAp, Ga-doped HAp exhibited a weakened osteoinductive potential. Furthermore, the study results indicated that GaOOH could also serve as a potential osteogenic material. This research provides valuable insights for the development of novel Ga-doped ceramic materials for bone repair applications.

## Data Availability

Not applicable.

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
