# Peer review of "Gallium-Doped Hydroxyapatite: Shape Transformation and Osteogenesis Activity"

_molecules, 2023, doi:10.3390/molecules28217379_

Round 1
Reviewer 1 Report
Comments and Suggestions for Authors
The paper reports on the preparation of gallium-doped hydroxyapatite and its evaluation as a osteogenic material. The subject of the paper fits the scope of Molecules journal.
I have following comments:
1. In the Introduction section, I would suggest adding relevant information on the use of various doped hydroxyapatites in bone tissue repair.
2. The incorporation of gallium ions into HAp crystal lattice is not confirmed by any experimental data. Instead, formation of amorphous Ga-containing phases on the surface of HAp particles can be supposed. To unambiguosly confirm Ga-doping (as it is claimed by the authors, e.g. in the title of the paper), additional experiments are needed. Please provide the information on X-ray diffraction maxima shift upon doping. Please also provide high-resolution EDX mapping data. Actually, it is hardly expected that 50% Ga-doping level can be theoretically achieved in HAp matrix.
3. Please correct chemical formulas in Lines 231, 233.
4. Please re-write Section 2.3 in the past tense.
5. Please indicate clearly whether GaOOH was used as a control in Figs. 6, 7.
Comments on the Quality of English Language
Moderate editing of English language is required
Reviewer 2 Report
Comments and Suggestions for Authors
1. Figure 6A, 20Ga and Hap groups are not significantly different. Should be labeled correctly. Mention what in the figure caption *. * is it a difference between the control and samples?
2. Figure 7 should be represented in different way to show how ALP, COL, and OC change with time. It’s not clear from the current figure.
Reviewer 3 Report
Comments and Suggestions for Authors
Thank you for your contributions:
I require a single image depicting the (1) detailed X-ray diffraction (XRD) patterns of the pure hydroxyapatite (HAP) structure, as well as a separate image displaying the XRD pattern of gallium (Ga).
This will enable me to critically analyze the peaks. If possible, please include these images on an additional page solely for my personal satisfaction, and not for inclusion in the paper.
(2) The aforementioned observation regarding FTIR also applies, as mentioned in the first comment.
(3) The methodology section is typically positioned before the results and discussion sections, relocate it as per journal format.
1. The main question addressed by the research is the potential of Ga-doped hydroxyapatite (HAp) nanoparticles as functionalized bone repair materials to accelerate the repair process of bone tissue defects in patients suffering from trauma, osteoporosis, and other pathological conditions.
2. The topic is relevant in the field of bone tissue regeneration and repair. It addresses a specific gap by exploring the use of gallium-doped HAp nanoparticles, which is a relatively novel approach to enhancing bone tissue repair. Gallium's multifunctional properties make it an interesting candidate for this purpose.
3. This research adds to the subject area by demonstrating that gallium can inhibit HAp crystal growth, promote the transformation of HAp nanoparticles, and positively influence the proliferation and osteogenic differentiation of bone marrow mesenchymal stem cells. These findings contribute valuable insights into the potential use of Ga-doped HAp in bone remodeling and repair compared to other published materials.
4. Regarding methodology, the authors should provide more details about the synthesis process of Ga-doped HAp nanoparticles. It would be helpful to include information on the specific concentrations of gallium used and the synthesis conditions. Further controls could involve additional in vitro and in vivo experiments to assess the long-term effects, biocompatibility, and mechanical properties of Ga-doped HAp.
5. The conclusions appear to be consistent with the evidence and arguments presented. The research demonstrates that Ga-doped HAp nanoparticles have the potential to promote bone tissue repair through various mechanisms, which aligns with the main question posed.
6. The references are appropriate.
7. The tables and figures are appropriate.
Reviewer 4 Report
Comments and Suggestions for Authors
-
it is unclear what is the novelty of the work compared with already known data; Literature review is also very modest;
-
Gallium is an analog of Al and is typical metal not a semimetal element (line 47). This is a mistake.
-
The purpose of the work is very unclear.
-
The abstract should contain only basic results, not description of the aim of the work, the the claim of the importance of the results (lines 11-16) and discussion of the literature data (lines 19-22);
-
the introduction is also written very poorly and does not meet general requirements. Gallium isn’t an essential element. According to recent investigations Ga is considered toxic to human body. How much of this element should be introduced into human body to reach “potential effects” (line 69)? It is claimed that nanosheet-like structures appear at 50% (!!!) doping of Ga. Is this concentration of Ga safe? Authors propose to use “highly concentrated Ga-doped HAp material” (lines 296-297).A choice of Ga has to be more distinctly justified taking into account a cumulative effect (lines 147-156) even if “10 Ga-HAp groups can promote OC gene expression” (lines 197-200).
English is quite understandable
